# Understanding the Expressive Capabilities of Knowledge Base Embeddings under Box Semantics

**Mena Leemhuis**    MENA.LEEMHUIS@UNIBZ.IT  and  **Oliver Kutz**    OLIVER.KUTZ@UNIBZ.IT
*Free University of Bozen-Bolzano, Italy*

**Editors:** Leilani H. Gilpin, Eleonora Giunchiglia, Pascal Hitzler, and Emile van Krieken

## Abstract

Knowledge base embeddings are a widely applied technique, used for instance to improve link prediction tasks on knowledge graphs by using the geometric regularities occurring during learning. Techniques where ontological concepts are interpreted as boxes have shown to be particularly useful in this context, as they are both suitably expressive and of low computational complexity. However, to use those regularities for learning, it is necessary to determine and understand the possible biases in the approach: how do we distinguish what is learned due to regularities in the data from what is simply based on the representational limitations of the embedding? In this paper, we establish that there are some severe limitations in expressivity when modeling description logic ontologies with box embeddings in intended target languages such as $\mathcal{ELHO}(\circ)^{\perp}$. We illustrate that, under some weak assumptions, box semantics always satisfy Helly's Property, and is thus too weak to semantically capture $\mathcal{ELHO}(\circ)^{\perp}$ in an adequate way. We then characterize how so-called Helly-satisfiable $\mathcal{ELHO}(\circ)^{\perp}$ ontologies can be determined. We discuss the implications of this result with respect to existing box embedding approaches and real-world use cases.

## 1. Introduction

*Knowledge Graphs (KGs)*(Hogan et al., 2021) are a widely used representation of diverse knowledge in form of $(subject, predicate, object)$-triples, e.g., $(alice, loves, bob)$. As KGs tend to be highly incomplete, it is necessary to predict missing triples. For this task, *Knowledge Graph Embedding (KGE)* has turned out to be useful, as it allows for using geometric regularities for learning. Though these approaches showed a promising result quality, they do not incorporate background knowledge. Several techniques have been proposed to include background knowledge in form of an ontology. Approaches are, e.g., based on sequence modeling, graph propagation and *Knowledge Base Embeddings (KBEs)* (see (Chen et al., 2025) for a survey). The basic idea of KBE is to model individuals as points in a geometric space, concepts as convex sets and relations and logical operations as geometric operations between the individuals or concepts. Subconcept relations are modeled as subset relations and an individual belongs to a concept if its representation is a member of the respective convex set, mimicking the set-based Tarskian semantics. This ensures that newly inferred triples adhere to the background knowledge. There are many different KBE approaches, varying in the choice of the representations of concepts and relations. For instance, they can be based on representing concepts as spheres (Kulmanov et al., 2019), closed convex cones (Özçep et al., 2020) or boxes (Peng et al., 2022; Xiong et al., 2022; Yang et al., 2025). To gain a trustworthy result for link prediction, it is necessary to determine whether the ontology has been modeled correctly. Additionally, it is necessary to ensure

that the embedding represents the geometric regularities of the training data, and not a bias imposed by possible restrictions of the embedding approach. This leads us to two questions that need to be answered for every KBE approach:

(1) Is the training procedure of the approach able to find an embedding where geometric regularities precisely reflect the information of the knowledge base?

(2) Does such an embedding always exist? If not, under what conditions does it exist?

We will focus here on the more general question (2), which is a basis for particular improvements in the training procedure considered in (1). These questions have been discussed for some specific KBE approaches, by Lacerda et al. (2024) in the context of the description logic $\mathcal{ELH}$ and convex sets and by Özçep et al. (2020) in the context of closed convex cones. Abboud et al. (2020) and Boratko et al. (2021) considered the expressivity of box embeddings, however, boxes were used to model relations and not concepts. Here, we are following the lines of Lacerda et al. (2024) but focusing on embeddings based on boxes. They are widely used as they exhibit a low computational cost and are able to represents various fragments of the description logic $\mathcal{ELHO}(\circ)^{\perp}$ (the exact expressivity varies for different approaches). However, their expressivity has not been thoroughly examined.

Though, Bourgaux et al. (2024) pointed out that box embedding approaches exhibit problems. For instance, some approaches are not able to model consequences of axioms. This means that whilst an axiom might hold in a geometric representation, its consequences might not necessarily be satisfied (thus it is not a full model of the knowledge base). These problems are, however, problems exhibited by *specific* box embedding approaches. We want to dig deeper into this problem and extend the work of Bourgaux et al. (2024) by understanding the general pattern. Assume we are given an *optimal* box embedding approach, based on some basic assumptions about box semantics. It is assumed to be optimal in the sense that, first, we do not consider its learnability but assume that the embedding is findable. Second, it only imposes restrictions that all box embedding approaches have. Is it then possible to model each $\mathcal{ELHO}(\circ)^{\perp}$-ontology such that the ontology is satisfiable if and only if there is a box model of it? In other words, are the limitations of current box embedding approaches based only on the specific (learning) approach used, or are these limitations based on general properties of box semantics? We show in the following that, in addition to restrictions imposed by specific box embedding techniques, also the second is the case. Thus it is in fact not possible to find a correct box embedding for each $\mathcal{ELHO}(\circ)^{\perp}$-ontology under some weak and widely accepted assumptions on box semantics.[1] This result is based on an analysis of *Helly's Property* (going back to Helly (1923)), a well-known fact about intersections of convex sets that can be applied to box semantics. Based on this property, we define the notion of *Helly-satisfiable ontologies*. We then illustrate the relevance of this result for real-world use cases, namely that it leads to unwanted inferences in the embedding space.

The paper is structured as follows: After discussing the preliminaries on description logics, boxes and box embeddings in Sec. 2, Sec. 3 discusses the expressivity of box embeddings. In Sec. 4, the implications of this result to real-world use cases are discussed. The paper ends with a short conclusion. The proofs can be found in the appendix.

---

1. These general standard assumptions on box embeddings, outlined in more detail below, include that conjunction is modeled as set-intersection and that the bottom concept is modeled as the empty set.

| Name | Syntax | Semantics |
|---|---|---|
| top | $\top$ | $\Delta$ |
| bottom | $\bot$ | $\varnothing$ |
| nominal | $\{a\}$ | $\{a^{\mathcal{I}}\}$ |
| conjunction | $C \sqcap D$ | $C^{\mathcal{I}} \cap D^{\mathcal{I}}$ |
| existential restriction | $\exists R.C$ | $\{x \in \Delta \mid \exists y \in \Delta : (x,y) \in R^{\mathcal{I}} \wedge y \in C^{\mathcal{I}}\}$ |
| role concatenation | $(R_1 \circ R_2)^{\mathcal{I}}$ | $\{(a,c) \mid \exists b \in \Delta : (a,b) \in R_1^{\mathcal{I}}, (b,c) \in R_2^{\mathcal{I}}\}$ |

Table 1: Syntax and semantics of $\mathcal{ELHO}(\circ)^{\bot}$ (Baader et al., 2005)

## 2. Preliminaries

In the following, description logics are introduced in Sec. 2.1, the definition of boxes in Sec. 2.2 and box embeddings in Sec. 2.3.

### 2.1. Description Logics

Ontologies are widely used to represent structured information of the world. One way of representing ontologies is with the help of *Description Logics (DL)* (Baader et al., 2007). We are focusing here on the $\mathcal{ELHO}(\circ)^{\bot}$-fragment of the well-known description logic $\mathcal{EL}^{++}$(Baader et al., 2005) due to its computational advantages, as subsumption is polynomial. Prominent examples for ontologies in $\mathcal{ELHO}(\circ)^{\bot}$ are, e.g., SNOMED (Donnelly, 2006) for clinical documentation and the Gene Ontology (Ashburner et al., 2000) for modeling genes.

A DL vocabulary is given by a set of individual names **I**, a set of role names **R** and concept names **C**. The $\mathcal{ELHO}(\circ)^{\bot}$ concepts over $\mathbf{C} \cup \mathbf{R}$ are described by the grammar

$$C \longrightarrow A \mid \{a\} \mid \bot \mid \top \mid C \sqcap C \mid \exists R.C$$

where $A \in \mathbf{C}$ is an atomic concept, $a \in \mathbf{I}$ is an individual name, $R \in \mathbf{R}$ is a role symbol, and $C$ stands for arbitrary concepts. $\{a\}$ denotes a nominal concept. An *ontology* $\mathcal{O}$ is defined as a pair $\mathcal{O} = (\mathcal{T}, \mathcal{A})$ of a *terminological box (Tbox)* $\mathcal{T}$ and an *assertional box (Abox)* $\mathcal{A}$. A Tbox consists of *general inclusion axioms (GCIs)* $C \sqsubseteq D$ ("$C$ is subsumed by $D$") with concept descriptions $C, D$, *role inclusions* $R_1 \circ \cdots \circ R_k \sqsubseteq R$ and *role hierarchies* $R_1 \sqsubseteq R$. Each ontology can be translated adhering to the following normal forms: all general concepts inclusions can be represented as follows (for $C, D \in \mathbf{C}, E \in \mathbf{C} \cup \{\bot\}$)

$$C \sqsubseteq E \qquad\qquad C \sqsubseteq \exists R.D \qquad\qquad C \sqcap D \sqsubseteq E \qquad\qquad \exists R.C \sqsubseteq E$$

and all role inclusions can be represented as $R_1 \sqsubseteq R$ or $R_1 \circ R_2 \sqsubseteq R$ for $R, R_1, R_2 \in \mathbf{R}$.

An Abox consists of a finite set of *assertions*, i.e., facts of the form $C(a)$ or of the form $R(a,b)$ for $a, b \in \mathbf{I}, C \in \mathbf{C}$ and $R \in \mathbf{R}$. An *interpretation* is a pair $(\Delta, \cdot^{\mathcal{I}})$ consisting of a set $\Delta$, called the *domain*, and an *interpretation function* $\cdot^{\mathcal{I}}$ which maps individual names to elements in $\Delta$, concept names to subsets of $\Delta$, and role names to subsets of $\Delta \times \Delta$. The semantics of arbitrary concept descriptions for a given interpretation $\mathcal{I}$ is given in Table 1. A concept inclusion $C \sqsubseteq D$ is represented as $C^{\mathcal{I}} \subseteq D^{\mathcal{I}}$, a role inclusion $R_1 \circ \cdots \circ R_k \sqsubseteq R$ as $R_1^{\mathcal{I}} \circ \cdots \circ R_k^{\mathcal{I}} \subseteq R^{\mathcal{I}}$. An interpretation $\mathcal{I}$ *models* an Abox axiom $C(a)$, for short $\mathcal{I} \vDash C(a)$, iff $a^{\mathcal{I}} \in C^{\mathcal{I}}$ and it models an Abox axiom of the form $R(a,b)$ iff $(a^{\mathcal{I}}, b^{\mathcal{I}}) \in R^{\mathcal{I}}$. An interpretation

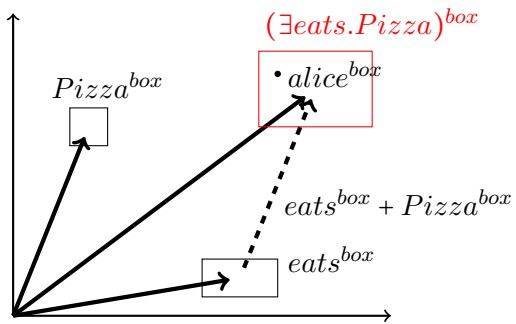
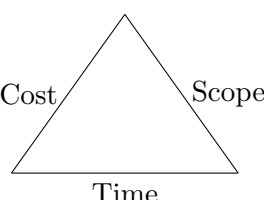

Figure 1: (a) Example for an embedding with TransBox; (b) project management triangle (Van Wyngaard et al., 2012)

is a *model of an ontology* $(\mathcal{T}, \mathcal{A})$ iff it models all axioms appearing in $\mathcal{T} \cup \mathcal{A}$. An ontology $\mathcal{O}$ *entails* a (Tbox or Abox) axiom $ax$, for short $\mathcal{O} \vDash ax$, iff all models of $\mathcal{O}$ are also models of $ax$. In this paper, the focus lies on finite ontologies.

### 2.2. Boxes

Boxes are chosen as a basis for many embeddings due to their good computational properties and simple representation. A box in some $\mathbb{R}^n$, for $n \in \mathbb{N}$, is defined as an axis-aligned hyperrectangle. It can be represented by its lower corner $l_c \in \mathbb{R}^n$ and upper corner $u_c \in \mathbb{R}^n$, with $l_c \leq u_c$, where $\leq$ is applied element wise. Then, $Box(C) = \{x \in \mathbb{R}^n \mid l_c \leq x \leq u_c\}$. Let $BoxHull(A)$ be the smallest box containing all elements of a set $A$. This can be defined as

$$BoxHull(A) = \{(x_1, \ldots, x_n)^T \mid x_i \in ConvHull(\{a_i \mid a \in A\}) \text{ for } 1 \leq i \leq n\}$$

where $ConvHull(X)$ is the convex hull of $X$ and $a_i$ is the value at the $i$-th dimension of vector $a$. The set $\mathcal{B}^n = \{BoxHull(X) \mid X \subseteq \mathbb{R}^n\}$, thus the set of all boxes in $\mathbb{R}^n$ including the whole space $\mathbb{R}^n$ and the empty set, is closed under set intersection. Properties of boxes are widely researched, e.g., in the context of *intersection graphs* and *boxicity* (Roberts, 1969).

### 2.3. Box Embeddings

KBE is a technique for using geometric regularities for learning purposes, especially for link prediction. It is based on the core idea of modeling concepts as convex regions in a low dimensional vector space. Then, individuals are interpreted either as points or as nominal concepts, and logical operations are modeled as some geometric operations in the space. For instance, the conjunction of two concepts can be represented as the intersection of the regions representing the respective conjuncts. We are focusing here on representing these convex sets as boxes, as they show a good tradeoff between expressivity and computational properties. Especially, they are closed under intersection (in contrast to spheres) and easy to be handled computationally (in contrast to cones). Box embedding approaches in the context of KBE are BoxEL (Xiong et al., 2022), ELBE (Peng et al., 2022), Box²EL (Jackermeier et al., 2024) and TransBox (Yang et al., 2025), which mainly differ in the representation of relations. Some represent individuals as points, some as nominals (thus as

small boxes), and TransBox interprets conjunction as approximation of intersection. They are considering $\mathcal{ELHO}(\circ)^{\perp}$ or a fragment of it. An example for a box embedding based on TransBox can be seen in Fig 1 (a). It is based on the idea of modeling relations also as boxes and states that a triple $(a, r, b)$ holds if $a \in box(r) + b$. In the example, the concept "Pizza" and the role "eats" are represented as boxes, the individual "alice" as a point in the space. Then it is the case that "Alice eats pizza" if the point representing Alice is part of the box representing $\exists eats.Pizza$. This box is determined by adding up the centers resp. the offsets of the boxes of "Pizza" and "eats".

Box embeddings are also of interest in other areas, e.g., probabilistic embeddings (Vilnis et al., 2018) and query embedding (Ren et al., 2020). Boxes are also used for modeling relations without considering concepts (Abboud et al., 2020).

As discussed by Bourgaux et al. (2024), KBE approaches need to fulfill the following properties: An embedding $\xi$ resulting of the application of a KBE approach is called *entailment closed* if every GCI, Abox and role assertion entailed by the ontology is also entailed by the embedding. This clearly implies that $\xi$ also satisfies $\mathcal{O}$. It is *weakly faithful* if the set of all GCI, Abox and role assertion that holds in the embedding model is consistent with the ontology. If these two are fulfilled, it remains to be shown that the existence of an embedding of $\mathcal{O}$ implies that $\mathcal{O}$ is satisfiable and that for every satisfiable ontology $\mathcal{O}$ an embedding can be found. Thus, the embedding should actually behave like a Tarskian model of the ontology and for each satisfiable ontology $\mathcal{O}$, it should be possible to find an embedding that is a model of $\mathcal{O}$. As argued by Bourgaux et al. (2024), these features are necessary in order to obtain an expressive and well-behaved embedding approach. However, they are *not* fulfilled for most of the existing box embedding approaches.

As the aim of this work is not to discuss the expressivity of one specific box embedding approach and as all of the existing approaches have some limitations regarding their expressivity, we present a generic box embedding approach as a basis for the examination of the expressivity. Based on some basic assumptions on the embedding, we show that it is not possible to express all $\mathcal{ELHO}(\circ)^{\perp}$-ontologies correctly, independently of the box embedding approach used. After considering those general limitations in the next section, we come back to the discussion of box embedding approaches in practice in Sec. 4.

## 3. The Expressivity of Box Embeddings

In the following, first a generalized box interpretation is presented that is used afterwards to discuss limitations of box embeddings.

### 3.1. A Generalized Box Interpretation

In the following, the general expressivity of box embeddings is considered. Therefore, we first determine basic properties of box embedding approaches and use them to define a common notion of box interpretation that is used as basis for further considerations. Classically, all concepts are interpreted as boxes: the top concept as the whole space $\mathbb{R}^n$, the bottom concept as the empty set, and other concepts as specific boxes. Furthermore, conjunction of concepts is typically defined as set-intersection, with the exception of TransBox (Yang et al., 2025) where an approximated intersection is considered. We use here the classical intersection. The main difference between the various approaches lies in the definition of

relations used. To define a well-behaved semantics for relations, we interpret them as sets over $\mathbb{R}^n \times \mathbb{R}^n$, however with the added condition that existential role restrictions always transform boxes into boxes. We arrive at the following definition.

**Definition 1** *A box interpretation $\xi$ is a structure $(\Xi, \cdot^\xi)$, where $\Xi = \mathbb{R}^n$ for some $n \in \mathbb{N}$, and where $\cdot^\xi$ maps each concept name $A \in \mathbf{C}$ to some box in $\mathcal{B}^n$, each individual name $c \in \mathbf{I}$ to a point $c^\xi \in \Xi$, each nominal concept $\{c\}$ to the box $\{c^\xi\}$, and each role $R \in \mathbf{R}$ to a subset $R^\xi \subseteq \Xi \times \Xi$ such that for every $B \in \mathcal{B}^n$: $R^{-1}(B) \in \mathcal{B}^n$. A box interpretation for arbitrary $\mathcal{ELHO}(\circ)^\perp$-concepts is defined recursively as*

$$(\top)^\xi = \Xi \qquad (\perp)^\xi = \varnothing \qquad (C \sqcap D)^\xi = C^\xi \cap D^\xi$$
$$(\exists R.C)^\xi = \{x \in \Xi \mid there\ is\ y \in \Xi\ with\ (x,y) \in R^\xi\ and\ y \in C^\xi\}$$
$$(R \circ S)^\xi = \{(a,c) \mid \exists b \in \Xi : (a,b) \in R^\xi, (b,c) \in S^\xi\}$$

*A box interpretation $\xi$ models an Abox axiom $C(a)$ for short $\xi \Vdash C(a)$ iff $a^\xi \in C^\xi$ and it models an Abox axiom of the form $R(a,b)$ iff $(a^\xi, b^\xi) \in R^\xi$.*

This interpretation is inspired by classical interpretations as defined in Table 1. It differs only in the fact that classical interpretations consider concepts as arbitrary sets whereas in a box interpretation each concept is represented as a box. The definition of the roles ensures that each $(\exists R.C)^\xi$ results in a box, independent of the choice of $C$.

**Example 1** *This box interpretation is general in the sense that it allows for interpreting some of the existing box embedding methods as special cases. Consider, e.g., TransBox and especially the example mentioned in Fig. 1 (a). There, individuals are defined as points in $\mathbb{R}^n$, concepts as boxes, and $\perp$ and $\top$ can be interpreted as the empty space and $\mathbb{R}^n$, resp. A direct translation of $R^{box}$ to $R^\xi$ is the following: $R^\xi = \{(a^\xi, b^\xi) \mid a^\xi \in R^{box} + b^\xi\ and\ b^\xi \in \Xi\}$. As $R^{box}$ is a box, also $R^{box} + b^\xi$ is a box. As translation with $R^{box}$ is linear, also each translation of an arbitrary box results in a box. With $\mathbf{C} = \{Pizza\}, \mathbf{R} = \{eats\}$ and $\mathbf{I} = \{alice\}$, this definition leads to the example mentioned in Fig. 1 (a).*

In contrast to other approaches, the box interpretations of Def. 1 can be interpreted as a special type of classical interpretation in the following sense:

**Proposition 2** *Let $\xi$ be a box interpretation of an $\mathcal{ELHO}(\circ)^\perp$-ontology $\mathcal{O}$ such that $\xi \Vdash \mathcal{O}$. Then (1) $\xi$ is entailment closed, (2) $\xi$ is weakly faithful, and (3) $\mathcal{O}$ is satisfiable in standard DL semantics.*

Note that KBE approaches are loss-based. Thus, the satisfaction of axioms is optimized in a step-wise fashion. Due to local minima, it is possible that an embedding that in fact satisfies the ontology is not found even though it exists. We assume here, for simplicity, that if such an embedding exists it can also be found.

In contrast to classical interpretations, the box interpretation allows for the notion of convexity and dimensionality. These geometric regularities ease the training but come to the prize of a restricted expressivity, as not every satisfiable $\mathcal{ELHO}(\circ)^\perp$-ontology has a box interpretation that satisfies $\mathcal{O}$: there is a satisfiable $\mathcal{ELHO}(\circ)^\perp$-ontology not having any box interpretation. The problem can be visualized with the following example:

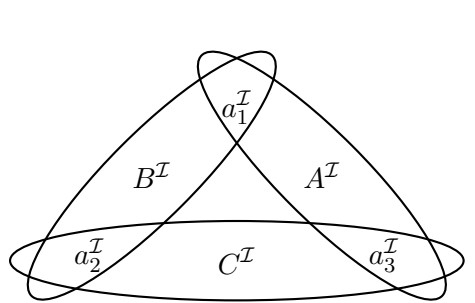 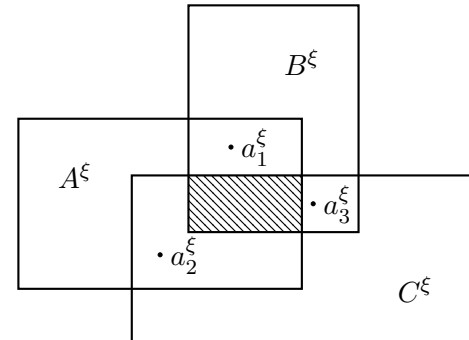

Figure 2: (a) a classical interpretation not fulfilling Helly's property; (b) a box interpretation not modeling $A \sqcap B \sqcap C = \bot$ (as the shaded region represents $A^\xi \cap B^\xi \cap C^\xi$)

**Example 2** *Given the ontology $\mathcal{O} = (\mathcal{T}, \mathcal{A})$ with $\mathcal{T} = \{A \sqcap B \sqcap C = \bot\}$ and $\mathcal{A} = \{A(a_1), B(a_1), B(a_2), C(a_2), C(a_3), A(a_3)\}$. $\mathcal{O}$ is satisfiable and a possible DL-interpretation can be seen in Fig. 2 (a). However, the attempt to find a model based on boxes leads to interpretations such as the one shown in Fig. 2 (b). With boxes, it is necessary to dismiss either the axiom $A \sqcap B \sqcap C = \bot$ or to model one of the individuals incorrectly.*

This restriction of boxes is in fact a classical result in the study of the properties of box intersections, called *Helly's Property (HP)*.

**Definition 3 (Helly's Property)** *(adapted from (Eckhoff, 1988))* A family $B$ fulfills Helly's Property *if it is the case that:* $\bigcap_{b \in B} b \neq \varnothing$ *if and only if for all $b_1, b_2 \in B$: $b_1 \cap b_2 \neq \varnothing$.*

**Proposition 4** *(adapted from (Eckhoff, 1988)) Each finite family $B \subset \mathcal{B}^n$ of axis-parallel boxes in $\mathbb{R}^n$ fulfills Helly's property, for any $n \in \mathbb{N}$.*

HP is a common and natural restriction, based on the well known *Helly's Theorem* (Helly, 1923) about the intersection of convex sets in $\mathbb{R}^n$ and can be found in many real world problems, e.g., in project management (Van Wyngaard et al., 2012). There, it is not possible to optimize production cost, production time and scope (thus, the number of features of the product) at the same time, however, each two of them can be optimized (see Fig. 1 (b)). It is, e.g., possible to produce a product cheap and fast, then, however, it is simple.

HP can be directly applied to box interpretations:

**Proposition 5** *There exists a classically satisfiable $\mathcal{ELHO}(\circ)^\bot$-ontology $\mathcal{O}$ such that no box interpretation in $\mathbb{R}^n$, for arbitrary $n \in \mathbb{N}$, satisfies $\mathcal{O}$.*

It is especially the case that this property is independent of the definition of the relations, as it also holds if $\mathbf{R} = \varnothing$, thus if relations are not considered at all. Thus the specific definition of the relations in Def. 1 does not influence this result.

This leads to several questions: (i) Is it possible to effectively check, for a given ontology $\mathcal{O}$, whether it has one (resp. has only) interpretation(s) fulfilling Helly's Property, thus whether it has a box interpretation satisfying $\mathcal{O}$? (ii) When it exists, can we effectively construct a Helly-satisfying interpretation? (iii) What is the influence on the embedding of real-world ontologies?

**Example 3** *First, question (iii) is considered. Many real-world ontologies such as GALEN (Rector et al., 1996) and SNOMED (Donnelly, 2006) are not modeling disjointness axioms at all and therefore always have a Helly-satisfiable model. However, there are still unwanted conjunctions that should be avoided. One especially fatal problem appearing due to Helly's property is the consideration of contraindications of drugs. Thus, there could be three drugs being all pairwise compatible having severe side effects given all together. Note that this argument is valid also for an empty Abox.*

## 3.2. $\mathcal{ELHO}(\circ)^{\perp}$ under HP-Semantics

There are two possible outcomes for checking an ontology in regard of HP. As proven in Prop. 5 an ontology could be not satisfiable by a box interpretation at all. Thus a learning approach for such an ontology leads necessarily to an erroneous result as explained in Ex. 2. Therefore, in such a case, either the ontology needs to be adapted, e.g., based on axiom weakening, or an approach not based on boxes needs to be considered. However, even if an ontology has a box interpretation fulfilling HP, it is not necessarily the most preferable one based on the geometric regularities. The box model is enforced to have certain properties and is therefore biased and less based on geometric regularities. Thus, a first idea would be to consider only ontologies as suitable for box embeddings of which every interpretation fulfills HP. This is, however, a severe restriction as depicted in the following example.

**Example 4 (An ontology having only interpretations fulfilling HP)** *Consider an ontology $\mathcal{O} = (\mathcal{T}, \mathcal{A})$ with $\mathcal{T} = \{\}$ and $\mathcal{A} = \{(A(a), B(a), B(b), C(c)\}$. It has a model not fulfilling HP as, e.g., the one in Fig. 2 (a). An ontology modeling this Abox and being restricted to having only interpretations fulfilling HP would either need a new individual d with $\{(A \sqcap B \sqcap C)(d)\} \in \mathcal{A}$ or an additional Tbox axiom, e.g., $\{B \sqcap C = \perp\} \in \mathcal{T}$ or $\{A \sqcap C = \perp\} \in \mathcal{T}$. Therefore, considering only ontologies satisfied only by HP-interpretations though would prevent from a bias, but would decrease the applicability of embedding approaches. The embedding would be restricted to ontologies where not too many inferences are possible. This would, however, contradict the main aim of KBE approaches.*

Therefore, we put up with the bias introduced due to the fact that only interpretations fulfilling HP can be represented. However, we want to find a restriction to circumvent cases where only an inconsistent interpretation can be found. Thus, we want to determine whether an ontology does not have an interpretation fulfilling HP at all. This is independent of the box interpretation but asks quite general on whether it is possible to find an adequate restriction for $\mathcal{ELHO}(\circ)^{\perp}$ so that there is at least one interpretation fulfilling Helly's property. This leads to the notion of *Helly-satisfiability*.

**Definition 6** *An $\mathcal{ELHO}(\circ)^{\perp}$-ontology $\mathcal{O}$ is* Helly-satisfiable *if it has a model $\mathcal{I}$ that fulfills Helly's property (Def. 3) for the set of definable concepts $DC_{\mathcal{O}}^{\mathcal{I}}$ in $\mathcal{I}$.*

$$DC_{\mathcal{O}}^{\mathcal{I}} = \{A \subseteq \Delta \mid \mathcal{I} \vDash \mathcal{O}, A = \varphi^{\mathcal{I}} \text{ for some } \varphi \in \mathcal{ELHO}(\circ)^{\perp}\}$$

Note that it is not sufficient to test HP for all representations of concept symbols but all concept representations need to be considered. Next, it is shown that if an ontology has an interpretation fulfilling HP, that it has a finite interpretation (thus it is constructable). This

is especially of importance, as the existence of an interpretation fulfilling HP is only useful if an embedding approach is (at least in theory) able to find it. First, note that Helly's property relies on the Abox-level: Consider an ontology $\mathcal{O} = (\mathcal{T}, \mathcal{A})$ with $\mathcal{T} = \{A \sqcap B \sqcap C = \bot\}$, $\mathcal{A} = \{\}$. It has an Helly-satisfiable interpretation, however, has the same Tbox as the ontology in Ex. 2. Therefore, it is not possible to define a rule on Tbox-level to capture this restriction. Therefore, an Abox closure rule is defined as an one-step closure procedure.

**Definition 7 (Helly-Abox closure rule)** *For all concept descriptions $A, B, C$: if $\mathcal{A}$ contains $\{A(a), B(a), B(b), C(b), A(c), C(c)\}$ but there is no individual $d$ with $\{A(d), B(d), C(d)\} \in \mathcal{A}$. Then, add a new individual $e$ and $\mathcal{A}' = \mathcal{A} \cup \{A(e), B(e), C(e)\}$.*

This rule basically checks whether there is a case where three concepts are pairwise intersecting. Then a new individual is added at the intersection of all three to circumvent contradictions to HP. If an ontology $\mathcal{O}$ is Helly-satisfiable, then with the help of this Abox closure rule, a finite model fulfilling HP can be found.

**Proposition 8** *Given a Helly-satisfiable $\mathcal{ELHO}(\circ)^{\perp}$-ontology $\mathcal{O}$, a finite model fulfilling HP can be found in finite time.*

Next we show that the Abox closure rule can be used to encode Helly-satisfiability in the notion of a Helly-companion, a materialization of HP in an extension of the ontology.

**Definition 9 (Helly-companion)** *An ontology $(\mathcal{T}', \mathcal{A}') = \mathcal{O}' \supseteq \mathcal{O} = (\mathcal{T}, \mathcal{A})$, with the sets $\mathcal{T}'$ and $\mathcal{A}'$ finite, is a Helly-companion of $\mathcal{O}$ if*

1. *$\mathbf{I}(\mathcal{O}') \supseteq \mathbf{I}(\mathcal{O}), \mathbf{C}(\mathcal{O}') = \mathbf{C}(\mathcal{O}), \mathbf{R}(\mathcal{O}') = \mathbf{R}(\mathcal{O})$; (Signature extends only ind. names)*

2. *If for some concept $C$ we have $\mathcal{O}' \cup \{C \sqsubseteq \bot\}$ is inconsistent, then there exists a $d \in \mathbf{I}(\mathcal{O}')$ such that $\mathcal{O}' \models C(d)$; (Every necessarily non-empty concept is witnessed.)*

3. *$\mathcal{A}'$ is Helly-closed for $\mathbf{I}(\mathcal{O}')$. (All Helly scenarios are witnessed.)*

Observe that the ontology constructed in the proof of Prop. 8 defines a satisfiable finite Helly-companion. Therefore:

**Proposition 10** *A satisfiable $\mathcal{ELHO}(\circ)^{\perp}$-ontology $\mathcal{O} = (\mathcal{T}, \mathcal{A})$ is Helly-satisfiable if and only if there exists a consistent Helly-companion $\mathcal{O}'$ of $\mathcal{O}$.*

This shows that it is (i) possible to determine whether an ontology is not Helly-satisfiable, thus whether there can't be a box embedding approach relying on the standard assumptions correctly modeling this ontology. (ii) it shows that if an ontology is Helly-satisfiable then a finite Helly-satisfiable model exists, thus there is (at least a theoretical) possibility to find this model. In the following, the influence of this result to box embeddings is discussed.

## 4. Helly in the Wild: Implications for Learning

The results of Sec. 3 show which types of $\mathcal{ELHO}(\circ)^{\perp}$-ontologies are embeddable using box interpretations. Learning approaches based on boxes either implicitly apply some sort of closure rules, as non-HP-interpretations are not representable, or first the Helly-companion

is determined and this is then embedded. However, then question (1) from the introduction comes into play: the existence of an interpretation does not imply that it is found in practice. This is based on two factors: first, the dimensionality of the embedding and second the increased complexity of the companion (or any interpretation fulfilling HP) in contrast to an arbitrary interpretation not necessarily satisfying $\mathcal{O}$. The complexity of the training procedure increases in higher dimensions and the curse of dimensionality especially complicates the modeling of intersections of boxes. Therefore, it is not possible to chose an arbitrarily high embedding dimension for learning. This is a severe restriction, as not all Helly-satisfiable $\mathcal{ELHO}(\circ)^\perp$-ontologies can be represented in a fixed dimension. This problem is heavily discussed in the context of boxicity (see, e.g., (Roberts, 1969)). Now, assume that the dimension of the embedding is appropriate and the learning approach actually finds a correct interpretation. Even then it is the case that only one possible interpretation of the ontology is modeled. Ideally, this interpretation is based on geometric regularities of the data, thus potentially newly inferred assertions are justified by the data. In box interpretations this is however not the case, due to HP: the interpretation fulfills HP due to the restrictions of box embeddings, not because the data suggests it. Therefore, it is especially of importance to determine the influence of HP to a specific ontology: which part of the embedding is solely based on the restriction of boxes to HP and which part is actually learned due to geometric regularities? Ideally, it should be possible to find an interpretation of the ontology such that all axioms and facts entailed by the interpretation are also entailed by the ontology. Thus, such an interpretation, in case it exists, can be considered 'the most general one'. The existence of such an interpretation can be read as implying a general ability for bias-free learning. For boxes, such a representation can only be found for a small fraction of ontologies (as discussed in Ex. 4).

## 5. Conclusion

KBE approaches are useful tools for inference tasks. However, a KBE approach is only usable if the bias of the approach is known, thus whether an inference is based on regularities in the data or restrictions of the model. We showed that KBE approaches based on boxes introduce a bias in form of Helly's property on the learned embedding. This bias both has influence on whether the interpretation is consistent and whether the inferences are based on geometric regularities. For future work, it is necessary to consider the dimensionality of interpretations further to determine whether an ontology can not only be modeled in theory but also in a restricted environment. Additionally, the faithfulness needs to be considered in more detail: is it possible to de-bias the embedding? Another interesting question is whether there are axioms that are preferably learned: for disjointness of concepts, it is, e.g., enough if the two respective boxes are disjoint in one dimension. In contrast, for non-disjointness, it is necessary that two boxes intersect in every dimension. However, there are use cases where it is appropriate to search for interpretations that only partially model the given ontology. Examples can include cases where the ontology is inconsistent, contains 'non-essential' axioms, or idiosyncratic individuals that could be omitted. Thus it will be essential to understand the deeper interplay between constraints imposed by the embedding semantics, restrictions imposed by the learning approach, and requirements imposed by the ontology languages.

## Acknowledgments

We acknowledge the financial support through the 'Abstractron' project funded by the Autonome Provinz Bozen - Südtirol (Autonomous Province of Bolzano/Bozen) through the Research Südtirol/Alto Adige 2022 Call.

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

## Appendix A. Proofs of Sec. 3.1

**Proof** [Proof of Proposition 2] Let $\mathcal{O}$ be an $\mathcal{ELHO}(\circ)^{\perp}$-ontology. Let $\xi \vDash \mathcal{T} \cup \mathcal{A}$. First, it is shown that each concept in $\xi$ actually is a box. Each concept symbol is interpreted as a box. $(\exists R.C)^{\xi}$ is defined as $R^{-1}(C^{\xi})$ and as $C^{\xi}$ is a box, by definition also $(\exists R.C)^{\xi}$ is a box. Boxes are closed under intersection. Therefore, also $(C \sqcap D)^{\xi}$ is a box for arbitrary concepts $C, D$. Note that $\varnothing \in \mathcal{B}^n$, thus by definition $(C \sqcap D)^{\xi} = \varnothing$ also results in a box. $\perp^{\xi}, \top^{\xi}$ and $\{c\}^{\xi}$ for nominals $\{c\}$ are boxes by definition.

In the following, it is shown that the box interpretation is a special case of a classical interpretation. As classical interpretations are entailment-closed and weakly faithful, the proposition follows.

Let $\mathcal{I}$ be a classical interpretation and let $\xi$ be a box interpretation that models all Tbox and Abox axioms. Let $\Delta = \Xi$, $c^{\mathcal{I}} = c^{\xi}$ for $c \in \mathbf{I}$. Let $C^{\mathcal{I}} = \{a \mid a \in C^{\xi}\}$ for $C \in \mathbf{C}$ and $R^{\mathcal{I}} = \{(a,b) \mid (a,b) \in R^{\xi}\}$ for $R \in \mathbf{R}$. Thus, $\mathcal{I} \vDash ax$ iff $\xi \Vdash ax$ for all assertions $ax$. Therefore, $\xi$ can be interpreted as classical interpretation and thus is entailment closed and weakly faithful. As $\mathcal{I}$ is a classical interpretation, (3) follows trivially. ∎

**Proof** [Proof of Prop. 5] Let $\mathbf{C} = \{A, B, C\}$ and $\mathbf{I} = \{a_1, a_2, a_3\}$ and let $\mathcal{T} = \{A \sqcap B \sqcap C = \perp\}$ and $\mathcal{A} = \{A(a_1), B(a_1), B(a_2), C(a_2), C(a_3), A(a_3)\}$.

Now, it is shown that for this ontology, it is not possible to construct a box interpretation satisfying $\mathcal{O}$. Assume by contradiction that such an interpretation exists in some $\mathbb{R}^n$. Thus, there are boxes $A^{\xi}$, $B^{\xi}$, $C^{\xi}$ and points $a_1^{\xi}, a_2^{\xi}, a_3^{\xi}$ with $A^{\xi} \cap B^{\xi} \neq \varnothing$, $A^{\xi} \cap C^{\xi} \neq \varnothing$, $B^{\xi} \cap C^{\xi} \neq \varnothing$, thus all elements of a set $B = \{A^{\xi}, B^{\xi}, C^{\xi}\}$ are pairwise intersecting, and therefore with Prop. 4 it follows that $A^{\xi} \cap B^{\xi} \cap C^{\xi} \neq \varnothing$, a contradiction. Therefore, $\xi$ does not satisfy $\mathcal{O}$, a contradiction. ∎

## Appendix B. Proofs of Sec. 3.2

The basic idea is to define an algorithm that constructs a Helly-companion by iteratively extending the Abox of the ontology with the concepts that need to be populated in every model of the ontology. This is done based on a construction principle similar to a tableau-algorithm. Therefore, first, the transformation rules are given. They are adapted from the

$\mathcal{ALC}$-tableau, see, e.g., (Baader and Sattler, 2001). Assume in the following for simplicity that the ontology is given in normal form.

**The $\to_\sqcap$-rule**
> **Condition**: $\mathcal{A}$ contains $(C_1 \sqcap C_2)(x)$, but not both $C_1(x)$ and $C_2(x)$.
> **Action**: $\mathcal{A}' := \mathcal{A} \cup \{C_1(x), C_2(x)\}$.

**The $\to_\sqsubseteq$-rule**
> **Condition**: $\mathcal{T}$ contains $C \sqsubseteq D$, $\mathcal{A}$ contains $C(x)$ (resp. $C_1(x)$ and $C_2(x)$ if $C = C_1 \sqcap C_2$), but not $D(x)$.
> **Action:** $\mathcal{A}' = \mathcal{A} \cup \{D(x)\}$.

**The $\to_{\exists\sqsubseteq}$-rule**
> **Condition**: $\mathcal{T}$ contains $\exists R.C \sqsubseteq D$, $\mathcal{A}$ contains $R(x,y), C(y)$, but not $D(x)$.
> **Action:** $\mathcal{A}' = \mathcal{A} \cup \{D(x)\}$.

**The $\to_{R \sqsubseteq S}$-rule**
> **Condition**: $\mathcal{T}$ contains $R \sqsubseteq S$, $\mathcal{A}$ contains $R(x,y)$, but not $S(x,y)$.
> **Action:** $\mathcal{A}' = \mathcal{A} \cup \{S(x,y)\}$.

**The $\to_\circ$-rule**
> **Condition**: $\mathcal{T}$ contains $R_1 \circ R_2 \sqsubseteq S$, $\mathcal{A}$ contains $R_1(x,y), R_2(y,z)$, but not $S(x,z)$.
> **Action:** $\mathcal{A}' = \mathcal{A} \cup \{S(x,z)\}$.

**The $\to_\exists$-rule**
> **Condition**: $\mathcal{A}$ contains $(\exists R.C)(x)$, but there is no individual name $z$ such that $C(z)$ and $R(x,z)$ are in $\mathcal{A}$.
> **Action**: $\mathcal{A}' := \mathcal{A} \cup \{C(y), R(x,y)\}$ where $y$ is an individual name not occurring in $\mathcal{A}$.

With these transformation rules, a tableau-inspired algorithm can be defined.

**Definition 11 (Adapted tableau algorithm for $\mathcal{ELHO}(\circ)^\perp$)** *Let $\mathcal{O} = (\mathcal{T}, \mathcal{A})$ be an $\mathcal{ELHO}(\circ)^\perp$-ontology. Apply the transformation rules iteratively to $\mathcal{A}$ by preferring all other rules over the $\to_\exists$-rule. The blocking condition is defined as usual (see (Baader and Sattler, 2001)): the application of the rule $\to_\exists$ to an individual $x$ is blocked by an individual $y$ in an Abox $\mathcal{A}$ iff $\{D \mid D(x) \in \mathcal{A}\} \sqsubseteq \{D' \mid D'(y) \in \mathcal{A}\}$.*

Based on classical tableau algorithms and due to the simplicity of $\mathcal{ELHO}(\circ)^\perp$, it can be proven that the algorithm terminates and that $\mathcal{O}' = (\mathcal{T}, \mathcal{A}')$ is satisfiable if $\mathcal{O}$ is satisfiable. The tableau algorithm allows for constructing a canonical interpretation.

**Definition 12 (Baader and Sattler (2001))** *The canonical interpretation $\mathcal{I}_\mathcal{A}$ of $\mathcal{A}$ is defined as follows:*

- *the domain $\Delta^{\mathcal{I}_\mathcal{A}}$ consists of the individual names occurring in $\mathcal{A}$*

- *for all concept names $P$ we define $P^{\mathcal{I}_\mathcal{A}} := \{x \mid P(x) \in \mathcal{A}\}$*

- *for all role names $R$ we define $R^{\mathcal{I}_\mathcal{A}} := \{(x,y) \mid R(x,y) \in \mathcal{A}\}$*

This algorithm is now extended by including the Abox closure rule of Definition 7. Therefore, first the application of the Abox closure rule is considered independently of the tableau. Observe that for a given interpretation $\mathcal{I}$ all concepts in $\mathcal{I}$ (thus $DC_{\mathcal{O}}^{\mathcal{I}}$) need to fulfill HP. Therefore, it is not sufficient to test only for concepts occurring in $\mathcal{A}$. Especially, it is necessary to construct a Helly-companion for $\mathcal{O}$ that fulfills HP. Special problems arise due to the consideration of roles: If $R(a,a) \in \mathcal{A}$, then it is necessary to test HP for each $\exists R.\top, \exists R.\exists R.\top$ etc. To circumvent this problem, a graph-based view is applied to the concepts.

**Definition 13** *Let $\mathcal{O} = (\mathcal{T}, \mathcal{A})$ be a $\mathcal{ELHO}(\circ)^{\perp}$-ontology. Let $a$ be an individual in $\mathcal{A}$. For $a$, a directed graph $G_a = (V_a, E_a)$ representing its relations is modeled as follows:*

$$V_a^0 = \{a_0\}$$
$$E_a^0 = \{(a_0, x) \mid R(a,x) \in \mathcal{A} \text{ for } R \in \mathbf{R}\}$$

*If $V_a^{i-1}, E_a^{i-1}$ are given, construct $V_a^i, E_a^i$ as follows:*

$$V_a^i = \bigcup\{y \mid \exists x : (x,y) \in E_a^{i-1} \text{ and } y \notin V_a^j \text{ for some } 0 \leq j < i \text{ and } R \in \mathbf{R}\}$$
$$E_a^i = \bigcup\{(x,y) \mid R(x,y) \in \mathcal{A} \text{ and } x \in V_a^i\}$$

*The procedure stops when $V_a^i = \varnothing$ thus no new nodes are added. Then $V_a = \bigcup_i V_a^i$, $E_a = \bigcup_i E_a^i$ This construction terminates, as $\mathbf{I}$ and $\mathcal{A}$ are finite and it is checked for duplicates.*

Note that such a graph can be directly translated into an assertion by considering the paths in the graph. For example let there be a $v \in V_a$ with $(a,v) \in E_a$ and $C \in \mathcal{C}(v)$ for $\mathcal{C}(v) = \{A \mid A(v) \in \mathcal{A}\} \cup \top$ (thus $\mathcal{C}(v)$ represents the concepts asserted to $v$ in the Abox). Then, $\mathcal{O} \vDash \exists R.C(a)$.

With the help of this graph, the Abox closure rule can be applied.

**Definition 14** *Let $\mathcal{O} = (\mathcal{T}, \mathcal{A})$ be a satisfiable $\mathcal{ELHO}(\circ)^{\perp}$-ontology. The Abox closure rule as defined in Def. 7 is applied to ontology $\mathcal{O}$ for all individual names $a, b, c$ occurring in $\mathcal{A}$ as follows:*

*First, create for each of $a, b, c$ the relation graph $G_a, G_b, G_c$ as defined in Def. 13. For each $v, w \in \{a, b, c\}$ now the combined relation graph $G_{v \cap w}$ is defined, thus the graph representing only concept representations asserted to both $v$ and $w$. Let $G_{v \cap w} = (V_{v \cap w}, E_{v \cap w})$.*

$$V_{v \cap w}^0 = \{\{v_0, w_0\}\}$$
$$E_{v \cap w}^0 = \{(\{v_0, w_0\}, \{x, y\}) \mid R(v,x), R(w,y) \in \mathcal{A} \text{ for } R \in \mathbf{R}\}$$

*The rest is defined analogously to Def. 13. Based on the same argument as above, $G_{v \cap w}$ is finite.*

*If for some $v, w$ $E_{v \cap w} = \varnothing$ and $\mathcal{C}(v) \cap \mathcal{C}(w) = \top$, then there is no non-trivial concept description $A$ with $A(v), A(w) \in \mathcal{A}$ and no $R \in \mathbf{R}$ with $R(v,x), R(w,y) \in \mathcal{A}$ for some individuals $x, y$. Thus, the closure rule is trivially fulfilled in this case.*

*Therefore, assume that for each of $v, w \in \{a, b, c\}$, $E_{v \cap w} \neq \varnothing$ or $\mathcal{C}(v) \cap \mathcal{C}(w) \supset \{\top\}$. Then, the premise of the Abox closure rule is non-trivially fulfilled and the conclusion needs to be tested and possibly a new individual needs to be added.*

*Thus, test whether there is an individual $d$ such that:*

- $(\mathcal{C}(a) \cap \mathcal{C}(b)) \cup (\mathcal{C}(b) \cap \mathcal{C}(c)) \cup (\mathcal{C}(a) \cap \mathcal{C}(c)) \subseteq \mathcal{C}(d)$ *(thus d shares all concepts that $a, b$ and $b, c$ and $a, c$ respectively share) and*

- *for each $v, w \in \{a, b, c\}$: Iteratively test for each edge and each node in $G_{v \cap w}$ whether an individual mimicking the modeled relation can be found. Therefore, construct the graph $G_d$ and find for each edge in $G_d$ matching edges in $G_{v \cap w}$. Start with edges $(d, z) \in E_d$. $(d, z)$ can be matched with $(\{v, w\}, \{x, y\}) \in E_{v \cap w}$, if $\mathcal{R}(v, x) \cap \mathcal{R}(w, y) \subseteq \mathcal{R}(d, z)$ where $\mathcal{R}(a, b) = \{R \mid R \in \mathbf{R} \text{ and } R(a, b) \in \mathcal{A}\}$ and if $\mathcal{C}(x) \cap \mathcal{C}(y) \subseteq \mathcal{C}(z)$. Thus, $(d, z)$ needs to have all relations that $(v, x)$ and $(w, y)$ share and $z$ needs to have all concepts that $x$ and $y$ share. This procedure is continued stepwise (thus, e.g., for edges $(z, u) \in E_d$ as match with all $(\{x, y\}, \{s, t\}) \in E_{v \cap w}$ for which in the last step a match has been found). If for all edges in $E_{v \cap w}$ a match is found, then $d$ has also a witness for each concept that $v$ and $w$ share.*

*If this is the case, then the Abox closure rule does not need to be applied. Otherwise, new individuals need to be defined Add a new individual $d_i$ for $0 < i \leq |V_{a \cap b}| + |V_{b \cap c}| + |V_{a \cap c}| - 3$ for each node in $G_{a \cap b}, G_{b \cap c}$ and $G_{a \cap c}$ except for the root nodes. For the root nodes add one individual $d_0$. For each $d_i$ add the corresponding concepts and roles to the Abox. For a $d_i$ corresponding to node $\{x, y\}$ of the graph, let $\mathcal{C}(d_i) = \mathcal{C}(x) \cap \mathcal{C}(y)$ and add $C(d_i)$ to $\mathcal{A}$ for all $C \in \mathcal{C}(d_i)$. For $d_0$ add $\bigsqcap_{A \in (\mathcal{C}(a) \cap \mathcal{C}(b)) \cup (\mathcal{C}(b) \cap \mathcal{C}(b)) \cup (\mathcal{C}(a) \cap \mathcal{C}(c))} A(d_0)$ to $\mathcal{A}$. For $i, j \geq 0$, add $R(d_i, d_j)$ to $\mathcal{A}$ if the corresponding $\{x, y\}, \{t, u\}$ have $R(x, t)$ and $R(y, u)$ in $\mathcal{A}$. Now, as new individuals have been added, the process is started again. This is repeated until nothing is added.*

Note that this construction is highly inefficient, as many individuals are added unnecessarily. These are, however, only finitely many and therefore not problematic as will be proven later on. This construction does not only apply the Abox closure rule but additionally considers the relational part. The relational part is, however, necessary to consider to gain an interpretation actually fulfilling HP. The process of Def. 14 is applied to all individuals in $\mathcal{A}$, thus also to the individuals newly added during the process. Therefore, it needs to be proven that this process terminates.

**Corollary 15** *The application of the (extended) Abox closure rule as stated in Def. 14 to an Abox $\mathcal{A}$ of an $\mathcal{ELHO}(\circ)^{\perp}$-ontology $\mathcal{O}$ terminates.*

**Proof**

1. First, consider the case where $E_{a \cap b} = \varnothing, E_{b \cap c} = \varnothing, E_{a \cap c} = \varnothing$ for all individuals $a, b, c$ considered during the process. In this case only concepts are considered. Assume $n$ is the number of different concepts occurring in $\mathcal{A}$, as $\mathcal{A}$ is finite by definition. Then, there are worst-case $2^n$ individuals to be added, as worst-case only $2^n$ different concept combinations are possible. If worst-case $2^n$ individuals have been added, then for each case when the premise of the Abox closure rule is fulfilled, there is an individual fulfilling the conclusion.

2. Now, consider the case where at least one of $E_{a \cap b} \neq \varnothing, E_{b \cap c} \neq \varnothing, E_{a \cap c} \neq \varnothing$ for some individuals $a, b, c$.

Observe the following facts: for each group of newly added individuals $d_i$, all of these elements except $d_0$ are not able to introduce new situations on which the Abox closure needs to be applied. This is the case, as each $d_i$ mimics the concepts represented in some subgraph of $G_{a \cap b}, G_{b \cap c}$ or $G_{a \cap c}$.

Thus, only $d_0$ needs to be considered. Note that $d_0$ does not have an incoming edge. Therefore, the same argument as for the case solely based on concepts can be used: There are finitely many graphs $G_v$, one for each individual in $\mathcal{A}$. There are only finitely many variants to combine these graphs. Therefore, only finitely many individuals can be added.

■

Until now, the process applies the Abox closure only to a given Abox.

As HP needs to be valid not only for the Abox but for an interpretation, it is additionally necessary to consider HP for concepts not present in the Abox but present in each possible interpretation satisfying an ontology. Thus, the Helly companion of an ontology needs to be defined (see Def. 9).

Therefore, in the following, the tableau algorithm is combined with the Abox closure rule to define a Helly-companion. Note that this is not the only possible Helly-companion. After that, it is shown that this approach terminates and leads to a model that fulfills HP.

**Definition 16** *In the following, a tableau algorithm incorporating the Abox closure rule is defined. Let $\mathcal{O} = (\mathcal{T}, \mathcal{A})$ be a satisfiable $\mathcal{ELHO}(\circ)^{\perp}$-ontology. Repeat the following two steps until the application of the Abox closure rule does not introduce any new individuals.*

1. *Apply the tableau algorithm as defined in Def. 11 on $\mathcal{O}$ until a blocking condition is reached. Then, materialize the blocking, thus add for each a blocked by b, for a the successors of b.*

2. *Apply the Abox closure as defined in Def. 14.*

Now, it is shown that this adapted tableau algorithm terminates.

**Corollary 17** *The algorithm as defined in Def. 16 terminates.*

**Proof** Each application of the tableau algorithm terminates and each application of the Abox closure terminates. Therefore, it remains to show that the combination of both terminates. It is again sufficient to consider the newly added $d_0$. All other added individuals represent concept descriptions that already exist in $\mathcal{A}$. If these would not be complete regarding the application of the transformation rules, then also the concepts used to construct the individuals would not have been complete.

However, for a new individual $d_0$ it can be the case that for $(A \sqcap B \sqcap C)(d_0)$ there is a Tbox axiom $A \sqcap B \sqcap C \sqsubseteq D$ (in its corresponding normal form). However, the Tbox is by definition finite. Therefore, there are only finitely many axioms of this type. Therefore, the algorithm terminates. ■

Now, it can be shown that this construction leads to a Helly-companion.

**Corollary 18** *Let $\mathcal{O} = (\mathcal{T}, \mathcal{A})$ be a satisfiable $\mathcal{ELHO}(\circ)^{\perp}$-ontology. Let $\mathcal{O}' = (\mathcal{T}', \mathcal{A}')$ be the result of the application of the modified tableau as defined in Def. 16. Then, $\mathcal{O}'$ is a Helly-companion.*

**Proof** Let $\mathcal{O} = (\mathcal{T}, \mathcal{A})$ be a satisfiable $\mathcal{ELHO}(\circ)^{\perp}$-ontology. Let $\mathcal{O}' = (\mathcal{T}', \mathcal{A}')$ be the result of the application of the modified tableau as defined in Def. 16. It is shown that $\mathcal{O}'$ is a Helly-companion.

1. $\mathcal{O} \subseteq \mathcal{O}'$, $\mathbf{I}(\mathcal{O}') \supseteq \mathbf{I}(\mathcal{O}), \mathbf{C}(\mathcal{O}') = \mathbf{C}(\mathcal{O}), \mathbf{R}(\mathcal{O}') = \mathbf{R}(\mathcal{O})$ follows trivially based on the definition.

2. Every necessarily non-empty concept is witnessed due to the transformation rules of the standard tableau and due to the fact that the tableau is sound and complete.

3. The process of Def. 16 terminates and directly models the Abox closure rule. Additionally, it due to the graph-based view, all complex concept descriptions including relations are considered and checked for HP. This means that $\mathcal{A}'$ is Helly-closed.

∎

**Corollary 19** *Let $\mathcal{O}$ be a Helly-satisfiable $\mathcal{ELHO}(\circ)^{\perp}$-ontology and let $\mathcal{O}'$ be the result of the application of Def. 16. The interpretation as defined in Def. 12 is a model of $\mathcal{O}$.*

**Proof** Let $\mathcal{O}$ be a Helly-satisfiable $\mathcal{ELHO}(\circ)^{\perp}$-ontology and let $\mathcal{O}'$ be the result of the application of Def. 16. Let $\mathcal{I}_{\mathcal{A}}$ be the canonic interpretation of $\mathcal{O}'$. Assume by contradiction that $\mathcal{I}_{\mathcal{A}}$ is not a model of $\mathcal{O}$. Therefore, there must be a $d^{\mathcal{I}_{\mathcal{A}}} \in (A \sqcap B \sqcap C)^{\mathcal{I}_{\mathcal{A}}}$ with $\mathcal{O} \vDash A \sqcap B \sqcap C = \perp$. When applying the standard tableau algorithm to a satisfiable ontology, a concept description $A$ is only added to $\mathcal{A}'$ if $\mathcal{O} \cup \{A \sqsubseteq \perp\}$ is inconsistent. Therefore, for all of these added concepts and same for all added roles, the Abox closure rule needs to be applied and thus HP needs to be tested. By definition, for fulfilling HP, always the least specific concept is added. Therefore such a $d^{\mathcal{I}_{\mathcal{A}}}$ would directly interfere with HP, thus, the ontology can't be Helly-satisfiable, a contradiction. ∎

**Corollary 20** *Let $\mathcal{O}$ be a Helly-satisfiable $\mathcal{ELHO}(\circ)^{\perp}$-ontology and let $\mathcal{O}'$ be the result of the application of Def. 16. The canonic interpretation $\mathcal{I}_{\mathcal{A}}$ of $\mathcal{O}'$ as defined in Def. 12 fulfills HP.*

**Proof** Let $\mathcal{O}$ be a Helly-satisfiable $\mathcal{ELHO}(\circ)^{\perp}$-ontology and let $\mathcal{O}'$ be the result of the application of Def. 16. $\mathcal{O}'$ is a Helly-companion of $\mathcal{O}$ (see Cor. 18). Therefore, it is Helly-closed. The canonic model of $\mathcal{O}'$ can be defined without the need to infer new conceptual information and without adding new relations or individuals. Therefore, the canonic model of $\mathcal{O}'$ is also Helly-closed and thus fulfills HP. ∎

Now, Prop. 8 can be proven. Thus, if an ontology is Helly-satisfiable, then there is a construction procedure for a finite model that satisfies HP.

**Proof** [proof of Proposition 8] Let $\mathcal{O} = (\mathcal{T}, \mathcal{A})$ be a Helly-satisfiable ontology in normal form.

Let $\mathcal{I}_\mathcal{A}$ be the canonic model of $\mathcal{O}'$ constructed based on $\mathcal{O}$ with the adapted tableau as defined in Def. 16.

$\mathcal{I}_\mathcal{A}$ is constructable in finite time, as proven in Cor. 17. It is actually a model of $\mathcal{O}$ due to Cor. 19 and it fulfills HP due to Cor. 20. Thus, a finite model of $\mathcal{O}$ fulfilling HP can be found in finite time. ∎

With this result, the proof of Proposition 10 follows.

**Proof** [Proof of Proposition 10]

→ Let $\mathcal{O} = (\mathcal{T}, \mathcal{A})$ be a Helly-satisfiable $\mathcal{ELHO}(\circ)^\perp$-ontology. Then apply the adapted tableau algorithm as defined in Def. 16 to get $\mathcal{O}'$ that is a Helly-companion as proven in Theorem 18. As proven in Cor. 19, a model can be defined, therefore, $\mathcal{O}'$ is satisfiable.

← Let $\mathcal{O}$ not be Helly-satisfiable. Consider an arbitrary Helly-companion $\mathcal{O}'$ of $\mathcal{O}$. It is shown that the resulting ontology $\mathcal{O}'$ is not satisfiable. As $\mathcal{O}$ is not Helly-satisfiable, there is in each model $\mathcal{I}$ of $\mathcal{O}$, $a^\mathcal{I}, b^\mathcal{I}, c^\mathcal{I} \in \Delta$ with $a^\mathcal{I} \in (A \sqcap B)^\mathcal{I}$, $b^\mathcal{I} \in (B \sqcap C)^\mathcal{I}$, $c^\mathcal{I} \in (A \sqcap C)^\mathcal{I}$ but $A^\mathcal{I} \cap B^\mathcal{I} \cap C^\mathcal{I} = \varnothing$ for some concept descriptions $A, B, C$.

Assume for the sake of contradiction that the Helly-companion is satisfiable. Each Helly-companion contains witnesses for each concept description that is known to be non-empty. Therefore, the canonic interpretation $\mathcal{I}_\mathcal{A}$ of $\mathcal{O}'$ would be a model of $\mathcal{O}'$ (as $\mathcal{O}'$ is assumed to be satisfiable). But this model fulfills by definition of the Helly-companion HP and therefore, $\mathcal{O}$ would be Helly-satisfiable. A contradiction to the assumption.

∎

