# OpenReview forum: "Understanding the Expressive Capabilities of Knowledge Base Embeddings under Box Semantics"
_nesyconf.org/NeSy/2025/Conference_Phase_2 — NeSy 2025 - Phase 2 Poster_

### Official Review · Reviewer_Lanu · 2025-06-29
**Good theoretical analysis on EL embedding methods. Discussion and conclusion require some work.**

**Rating:** 6
**Confidence:** 4

**Review:**

* Main comment:
The paper explores theoretical limitations of Box Embeddings of ELHO
ontologies and provides theoretical results regarding the
intersection operation in box-based models. The paper is quite
relevant to the KRR community.

* Comments:

  - In Example 4 there is ABox $\mathcal{A}$ which has a model in Fig 1
  (a). However, Fig 1 (a) seems unrelated. Also, from the ABox
  $\mathcal{A}$ I understand that $A \sqcap B \sqcap C \sqsubseteq
  \bot$. So, why adding $B \sqcap C \sqsubseteq \bot$ would suffice to
  fulfill the HP property? I think I am missing something here because I
  understand that the HP property would require A and B to be
  disjoint. I'd appreciate if you can clarify this example.

  - Many real world and widely used ontologies such as the Gene Ontology
  or the Human Phenotype Ontology have an empty ABox and disjointness
  of concepts are encoded in the TBox. These ontologies have been used
  in the mentioned embedding methods. I wonder what would be the
  implication of Helly Property in this type of ontologies. Is it
  possible to provide a closure rule or to adapt the Helly-Abox
  closure rule?

  - Section 4 is somewhat vague and hard to follow. For example, the
  authors spend half of the section discussing embedding dimension to
  finally conclude that applying the HP will guide the model to find a
  different interpretation than not applying the HP. Also, I am not
  sure that Section 4 is the best place to put the definition of
  "Strong Faithfulness" maybe this can be moved to the Preliminaries
  section. The last sentence is also quite vague: What other factors
  would be involved?  Why do you mention faithful representations? How
  does faithful representations relate to the HP property?

  - As a follow up in my last comment, I wonder how the HP property can
  affect learning if the ABox closure rule is applied? For example, in
  the cases where ontologies with empty ABox that are widely used such
  as GO?

* Minor comments
  - the embedding is finable --> the embedding is findable?
  - if [...], that is has a finite representation --> if [...], then it
  has a finite representation?

**Anonymity:**

Remain anonymous

---

### Official Review · Reviewer_wc4R · 2025-07-07
**Solid theoretical contribution on limitations of geometric ontology embeddings**

**Rating:** 7
**Confidence:** 3

**Review:**

The paper investigates the expressive limitations of box embeddings for ELHO(○) ontologies, showing that due to Helly’s Property, some satisfiable ontologies cannot be represented using axis-aligned boxes. It formalizes the notion of Helly-satisfiability and provides a method to check whether an ontology admits a box-based interpretation.

Pros:
- Very interesting work regarding the expressiveness of box embeddings in ontology modeling; The paper theoretically proves that box embeddings inherently satisfy Helly’s Property, limiting their ability to represent certain ELHO(o) ontologies;
- Results are not tied to any specific embedding method, making the findings broadly relevant across box-based KBE approaches;
- Definitions, propositions, and proofs are precise and rigorous, demonstrating solid theoretical grounding.

Cons:
- The paper feels very heavy with notation and terminology, which clutters the reading experience. That said, it seems like there’s plenty of room to make it more readable with some restructuring;
- The theoretical framework is well-constructed, and the limitations of box embeddings are carefully analyzed. However, the work would benefit from empirical validation (e.g., evaluating whether existing methods fail to find Helly-satisfiable embeddings in practice due to dimensionality or optimization);
- The paper broadly refers to "classical interpretations", but it would have benefited from a clearer connection to how existing embedding methods relate to or approximate these interpretations in practice;
- While the work is theoretically significant, its relevance may be limited to a relatively niche community interested in the formal properties of box embedding models.


Here's some additional comments:


- "and is thus too weak to capture semantically ELHO(o) in an adequate way" -> "semantically capture"

- "mimicking the set-based Tarskian semantics .." -> add a reference for the unfamiliar reader.

- Please rephrase question (1) in the introduction "Is the training procedure able to find an embedding where geometric regularities precisely reflect the information of the knowledge base?". It reads way to vague, what "training procedure"? Of a particular embedding model? A box-based one?

- "they show a severe increase in computational complexity (closed convex cones)" can the authors further support this claim?

- "An embedding ξ is called entailment closed if every GCI, Abox and role assertion entailed by the ontology is also entailed by the embedding": It might be worth clarifying what exactly is meant by “embedding” in this context, as some readers might initially interpret ξ as a vector embedding rather than something closer to a model or interpretation. A slightly more specific term like “embedding model” or "mapping" could help.

- "This interpretation is inspired by classical interpretations.": can the authors provide some examples?

- It might strengthen the paper to include a schematic or table comparing how different embedding approaches (e.g., BoxEL, ELBE, TransBox) model concepts, roles, and logical operators. This would help clarify which assumptions are truly general across methods and which are approach-specific.

- Rephrase "(ii) it shows that if an ontology is Helly-satisfiable that then a finite Helly-satisfiable model exists, thus that there is (at least a theoretical) possibility to find this model." -> (ii) it shows that if an ontology is Helly-satisfiable then a finite Helly-satisfiable model exists, thus there is (at least a theoretical) possibility to find this model.

This paper provides a strong theoretical analysis of the expressivity limits of box embeddings for ELHO(o) ontologies, showing that Helly’s Property imposes an inherent geometric constraint that prevents full coverage of logically satisfiable ontologies. The results are general, well-formalized, and go beyond specific implementations, making them broadly relevant to the knowledge base embedding community. As someone who has worked with box embeddings and ontologies to some extent, I would have really liked to have this kind of resource in literature. Therefore, I am leaning towards acceptance.

**Anonymity:**

Remain anonymous

---

### Official Review · Reviewer_3pNJ · 2025-07-10
**a theoretical analysis of the properties of box embeddings**

**Rating:** 6
**Confidence:** 3

**Review:**

The paper presents a theoretical analysis of the properties of box embeddings and whether they can faithfully capture ​​ELHO(○)*  ontologies. It is focused on discussion one example, where the pairwise intersection of classes is considered, to highlight that the empty three-way intersection is not capturable by the box embeddings. The paper is relevant for NeSy

The paper does not mention other types of KGE embeddings and thus appears to say that only geometric models are used to capture ontologies. A brief contextualization is needed.

The paper could be improved by providing a clearer example thatnthe project management triangle
It is not immediately obvious, and the figure, by having four labels and not three, is not helping.

It would be nice to have some examples of real world ontologies that use this language.

The paper is mostly well-written, but there are some agrammatical sentences and typos (see minor issue).

I do not find the paper particularly interesting. It boils down to the case of showing that box embeddings have a limitation: they assume that three paired pairwise intersections imply a three-way intersection. Why this is particularly useful is never clearly motivated. Moreover, it would be good to have a clear demarcation of contributions over Borgaux 2024.


Absolutely—here’s an improved, polished version of your review text:

The paper presents a theoretical analysis of the properties of box embeddings and whether they can faithfully capture ELHO(∘)| ontologies, placing it on topic for NeSy. It focuses on discussing a specific example in which the pairwise intersections of classes are considered to illustrate that an empty three-way intersection cannot be represented by box embeddings.
However, the paper does not discuss other types of embedding models and therefore gives the impression that only geometric embeddings are applicable for representing ontologies. A brief contextualization of alternative approaches would strengthen the paper.
The example used is an analogy to the project management triangle, and it is not immediately obvious. The accompanying figure, which includes four labels rather than three, further detracts from its clarity. A more direct or simplified example would improve reader comprehension.
It would also be helpful to include examples of real-world ontologies that fall on this fragment of DL, to better motivate the practical relevance of the work.
Overall, the paper is mostly well-written, though there are a few ungrammatical sentences and typos (see minor issues).
I did not find the paper particularly engaging, as it presents a rather narrow point: that box embeddings implicitly assume that three pairwise intersections entail a non-empty three-way intersection. The significance of this is not clearly articulated. Moreover, the contributions would benefit from a clearer delineation relative to Borgaux (2024).

MINOR ISSUES:
Examples of sentences that should be improved:
“Now, first a generalized box interpretation is presented”
“that it allows for interpretation some of the existing”
“whether it has one (has only) interpretation(s) fulfilling Helly’s Property”.
“there are axioms that are “easier” learnable”

Page 2: it's consequences -> its consequences

“either the ontology needs to be adapted (if possible)”, -> this sentence has two conflicting interpretations: (1) that the solution is to change the ontology (which is not what the authors mean, I believe) and (2) that it may not be possible to adapt the ontology. Please clarify.

**Anonymity:**

Remain anonymous